# Personalized education approach based on cognitive psychology for endoscopic diagnosis: A multicenter randomized trial

Kentaro Mochida[1,2], Fumiaki Ishibashi[1]*, Masamichi Yuzawa[3], Daisuke Suto[4], Toshihiro Nishizawa[5], Yasunari Sakamoto[6], Mikinori Kataoka[7], Mitsuru Esaki[8], Chika Kusano[9], Takuji Gotoda[10,11], Kenichi Konda[12], Jun Arimoto[13], Ryu Tanaka[14], Kohei Yamanouchi[15], Masao Okubo[16], Kazuma Fujimoto[2], Tomohiro Kawakami[17], Mizuki Nagai[1], Sho Suzuki[1]

1 Department of Gastroenterology, International University of Health and Welfare Ichikawa Hospital, Chiba, Japan, 2 International University of Health and Welfare Graduate School of Medicine, Tokyo, Japan, 3 Hiroshima University, Graduate School of Humanities and Social Sciences, Hiroshima, Japan, 4 Department of Gastroenterology, International University of Health and Welfare Hospital, Tochigi, Japan, 5 Department of Gastroenterology and Hepatology, International University of Health and Welfare Narita Hospital, Chiba, Japan, 6 Department of Gastroenterology and Hepatology, International University of Health and Welfare Atami Hospital, Shizuoka, Japan, 7 Department of Gastroenterology, International University of Health and Welfare Mita Hospital, Tokyo, Japan, 8 Department of Medicine and Bioregulatory Science, Graduate School of Medical Sciences, Kyushu University, Fukuoka, Japan, 9 Department of Gastroenterology, Kitasato University School of Medicine, Kanagawa, Japan, 10 Division of Gastroenterology and Hepatology, Department of Medicine, Nihon University School of Medicine, Tokyo, Japan, 11 Department of Gastroenterology, Cancer Institute Hospital, Tokyo, Japan, 12 Department of Medicine, Division of Gastroenterology, Showa University School of Medicine, Tokyo, Japan, 13 Department of Gastroenterology, Omori Red Cross Hospital, Tokyo, Japan, 14 Digestive Diseases Center, Shinjuku Tsurukame Clinic, Tokyo, Japan, 15 Department of Gastroenterology, Takagi Hospital, Fukuoka, Japan, 16 Department of Gastroenterology, Sanno Hospital, Tokyo, Japan, 17 Endoscopy Center, Koganei Tsurukame Clinic, Tokyo, Japan

* ishibashi-gast@iuhw.ac.jp

## Abstract

The learning curve for endoscopic diagnosis varies, and an optimal educational strategy is not available. Working memory (WM) refers to an individual's ability to store and process information quickly and simultaneously. This study aimed to determine whether educational methods, optimized by individual WM, facilitate learning about endoscopic diagnoses. In the development phase, the standard WM profile for general endoscopists was determined by 79 endoscopists. In the validation phase, 60 trainees from four Japanese institutions were assessed for visuospatial or verbal WM dominance, based on the standard WM profile, and randomly assigned to receive matched (Matched-E group) or unmatched (Unmatched-E group) education. In the Matched-E group, the visuospatial and verbal WM-dominant trainees learned the endoscopic diagnosis of colorectal polyps by pattern recognition and through the description text of narrow-band imaging classification, respectively. In the Unmatched-E group, participants received education that was opposite to their dominant WM type. The diagnostic accuracy of the endoscopic diagnosis of

**Data availability statement:** All data files are available from the UMIN database (accession number UMIN000050138) (URL: https://www.umin.ac.jp/ctr/index-j.htm).

**Funding:** This work was supported by International University of Health and Welfare Research Funds for Academic Year 2023. The funders had no role in study design, data collection and analysis, decision to publish, or preparation of the manuscript.

**Competing interests:** FI received honoraria for lectures from the Fujifilm Corporation. ME is a consultant for AI Medical Service, Inc. CK received honoraria for lectures from the Fujifilm Corporation and Olympus Corporation. TG received honoraria for lectures from the Fujifilm Corporation, Fujifilm Medical, and Olympus Corporation. SS received honoraria for lectures from the Fujifilm Corporation and Olympus Corporation. The authors declare no conflicts of interest.This does not alter our adherence to PLOS ONE policies on sharing data and materials.

colorectal polyps was compared between the groups after each educational session. Among the 60 trainees, 40 (21 in Matched-E and 19 in Unmatched-E) completed the validation test. The diagnostic accuracy was significantly higher in the Matched-E group than that in the Unmatched-E group (61.6% vs. 53.9%, P = 0.008). The diagnostic accuracy for non-neoplastic lesions was higher in the Matched-E group than that in the Unmatched-E group (68.0% vs. 48.4%, P = 0.002), whereas it did not differ for adenoma, intramucosal cancer, or invasive cancer. The personalized education based on the WM profile facilitated learning endoscopic diagnosis.

## Introduction

Colorectal cancer is the second leading cause of cancer-related deaths worldwide [1], and the removal of adenomatous polyps prevents death from colorectal cancer [2]. Consequently, high-quality endoscopic examinations, in addition to adequate detection, appropriate diagnosis, and efficient treatment of colorectal neoplasms, are required in daily practice [3,4]. Furthermore, accurate diagnostic skills can reduce unnecessary treatments and minimize the disadvantages associated with endoscopic examinations and treatment [5].

Several studies have reported that education based on classification systems, such as narrow-band imaging (NBI), facilitates the acquisition of diagnostic skills for colorectal polyps [6,7]. However, the acquired diagnostic accuracy varies among trainees, despite the use of the same educational method for unified diagnostic classification [8]. Additionally, feedback on endoscopy performance has yielded different results among trainees [9]. This discrepancy in the effectiveness of uniform training and feedback among trainees may be because of differences in their characteristics [9]. Therefore, personalized education may improve the abilities of trainees to learn endoscopy. However, no established personalized or tailored educational methods are available for endoscopy.

Cognitive psychology aims to elucidate the functions of cognition in the human mind, particularly in terms of perception, memory, thinking, language, learning, decision making, and behavioral selection. Recently, it has expanded its reach to the peripheral fields of psychology and a wide range of academic fields, such as philosophy, engineering, medicine, and art. Working memory (WM) is a cognitive psychological concept that refers to the ability to simultaneously process and store information in the brain over a short period of time [10]. The capacity of the WM directly affects its performance in various activities, including learning. Therefore, profiling individual WM is used to develop tailored educational methods [11,12]. An attempt to develop education based on WM profiles has revealed that two types of WM are important in characterizing the WM profile of an individual: visuospatial and verbal WM [13]. In particular, learning strengths differ depending on whether visuospatial or verbal WM is dominant. Consequently, previous studies have demonstrated a method for assessing WM profiles in a test format [14,15]. Additionally, we have developed a web-based test platform in Japanese (Hiroshima University Computer-Based Rating

of Working Memory [HUCRoW]) to measure individual WM [16]. Accordingly, in this study, we aimed to determine the standard WM profile for general endoscopists and evaluate the efficacy of personalized education, based on individual WM types, for the endoscopic diagnosis of colorectal polyps.

## Materials and methods

### Study setting

The evaluation of educational programs based on individual WM profiles consisted of two phases: development and validation (Fig 1). In the development phase, a standard WM profile for Japanese endoscopists was determined by collecting WM profiles from endoscopists from various backgrounds. In the validation phase, the effectiveness of personalized education programs was verified in a randomized controlled trial (RCT) of trainees whose WM was assessed using the standard WM profile determined during the development phase. This study was approved by the Institutional Review Board of the International University of Health and Welfare and each participating institution on January 11, 2023 (approval number: 22-Im-036). This study was registered with the University Hospital Medical Information Network Clinical Trials Registry (UMIN-CTR) (registration number: UMIN000050138). The study was conducted between January 2023 and January 2024, in accordance with the Declaration of Helsinki. Written informed consent was obtained from all study participants prior to their enrollment. All participants were adults aged 18 years or older who voluntarily agreed to take part in the study.

Overall, 79 endoscopists answered the WM assessment test to define the standard WM profile in the development phase. Subsequently, 60 trainees were enrolled in the validation phase, and their WM profile was assessed. Among them, 39 were visuospatial WM-dominant and 21 were verbal WM-dominant. The trainees were randomly allocated to two groups: 31 in the matched education group (Matched-E group) and 29 in the Unmatched-E group (Unmatched-E group). During the study, six trainees did not complete the education, four did not complete the final diagnosis test in the Matched-E group, and 10 did not complete the education in the Unmatched-E group. After excluding these participants, 21 and 19 participants in the Matched-E and Unmatched-E groups, respectively, were subjected to the final analysis.

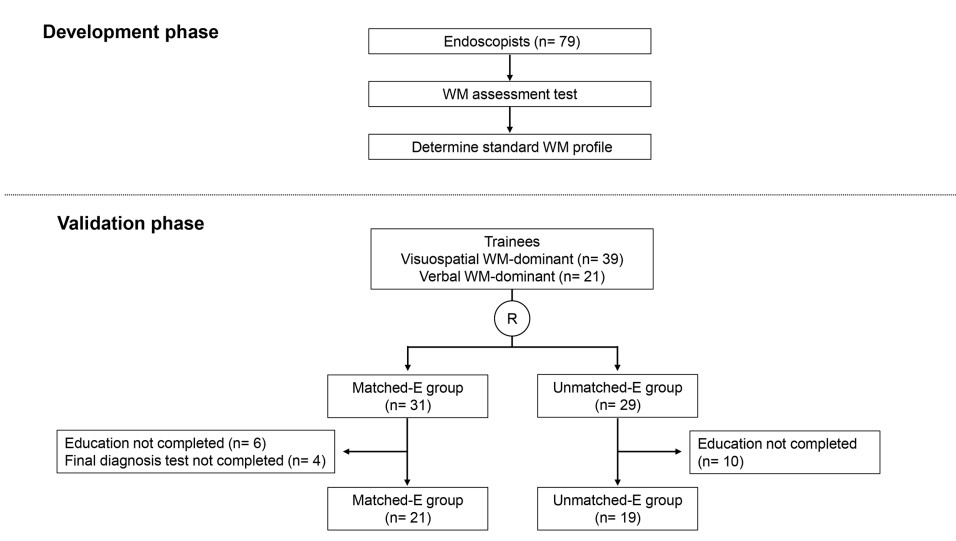

**Fig 1. Study flow diagram.**

### Development of an endoscopist's standard WM profile

Endoscopists from 13 institutions in Japan were recruited between April and July 2023, and their WM profiles were assessed using a web-based WM assessment test platform (HUCRoW). The HUCRoW consists of two tasks: the comparative line span (visuospatial WM) and backward digit span (verbal WM) (S1 and S2 Fig). The endoscopists' average scores and standard deviations were calculated for these tasks and were considered the standard WM profile for endoscopists.

### Validation of optimized education based on WM profiles

An RCT was conducted to evaluate personalized education based on the WM profile for trainees to learn about endoscopic diagnoses. Trainees at four institutions (International University of Health and Welfare Narita Hospital, Mita Hospital, Atami Hospital, and International University of Health and Welfare Hospital), with less than 2 years of experience and who had never performed colonoscopies, were enrolled between November and December 2023. Prior to randomization, all participants underwent a WM assessment test, similar to the one in the development phase. We determined the dominant WM (visuospatial or verbal) for each participant based on their scores in the two tasks. The deviation in each participant's scores on the two tasks was calculated with reference to the standard WM profile determined during the developmental phase. The WM type, either visual or verbal, with the highest deviation was defined as the dominant WM for each participant.

### Randomization

Subsequently, the participants were randomly assigned to the Matched-E group or Unmatched-E group in a 1:1 allocation ratio (Fig 1). The education group was randomly assigned by the UMIN Research Support INDICE Cloud System via dynamic balancing using the minimization method. A computer-generated randomization list was used, which was pre-ordered for each stratum. All investigators concealed the randomization sequence. The participants and investigators were blinded to the allocation of the observation group.

### Intervention

We developed two educational programs based on visuospatial and verbal characteristics to educate trainees about endoscopic diagnosis. The visuospatial education program was a pattern-recognition program that linked endoscopic images with a pathological diagnosis of 50 colorectal polyps (22 adenomas, seven invasive cancers, and 21 hyperplastic polyps). The verbal education program was used to learn the descriptive text of the surface structure and vascular patterns of 50 colorectal polyps according to the NBI International Colorectal Endoscopic Classification (NICE classification) [17] (S1 Table) without using endoscopic images.

The Matched-E group received training suited to their dominant WM type: visuospatial WM-dominant trainees learned through the visuospatial education program, and verbal WM-dominant trainees learned through the verbal education program. In contrast, the Unmatched-E group received education opposite to their dominant WM type: visuospatial WM-dominant trainees learned through the verbal education program, and verbal WM-dominant trainees learned through the visuospatial education program. Both education programs were available to the participants online. The learning time was set to 10 min. The participants could repeat the learning program as often as they wished within the time limit.

### Study endpoints

The primary endpoint was the diagnostic accuracy of colorectal polyps after each education session between the Matched-E and Unmatched-E groups.

$$Diagnostic\ accuracy\ (\%) = \frac{The\ number\ of\ correct\ diagnosis}{The\ number\ of\ lesions} \times 100$$

After each educational program, the participants diagnosed 50 colorectal polyps from the following three possible pathological diagnoses: non-neoplastic lesion, adenoma/intramucosal cancer, and invasive cancer. The pathological diagnoses of the 50 lesions in the test consisted of 21 non-neoplastic lesions, 22 adenomas or intramucosal cancers, and seven invasive cancers.

The secondary endpoints were diagnostic accuracy according to the participant's sex, years of experience as a doctor, lesion pathology, and response time to answer the questions. Additionally, the diagnostic accuracy according to the lesion pathology was evaluated to determine the ability to distinguish each pathological category from the other two.

## Sample size calculation and statistics

Education according to the NICE classification previously improved diagnostic accuracy by approximately 8–20% for trainees [8]. Therefore, we assumed that the diagnostic accuracy for colorectal polyps could improve by 20% and 10% after receiving matched and unmatched education for the WM, respectively. Consequently, using a standard deviation of 10%, a significance level (α) of 0.05, and a power (1-β) of 0.8, a minimum sample size of 34 was determined to be necessary for endpoint evaluation. Considering the possibility of dropping out (because of incomplete WM assessment, assigned education, or endpoint test), 40 participants were required.

All statistical analyses and sample size calculations were performed using R software (version 4.0.4). The t-test was used to compare continuous variables, whereas the chi-squared test or Fisher's exact test was used to compare categorical variables. Statistical significance was set at $P < 0.05$.

## Results

### Standard WM profile

Overall, 79 endoscopists with various backgrounds and characteristics performed the WM assessment tests that determined the standard WM profiles (S2 Table). The score distribution is shown in Fig 2. The average scores for the 79 endoscopists were 20.4±6.2 for the comparative line span assessing visuospatial WM, and 35.0±7.9 for the backward digit span assessing verbal WM. The average scores were considered to correspond to a deviation of 50. The difference in deviation between visuospatial and verbal WMs varied widely among endoscopists (S3 Fig).

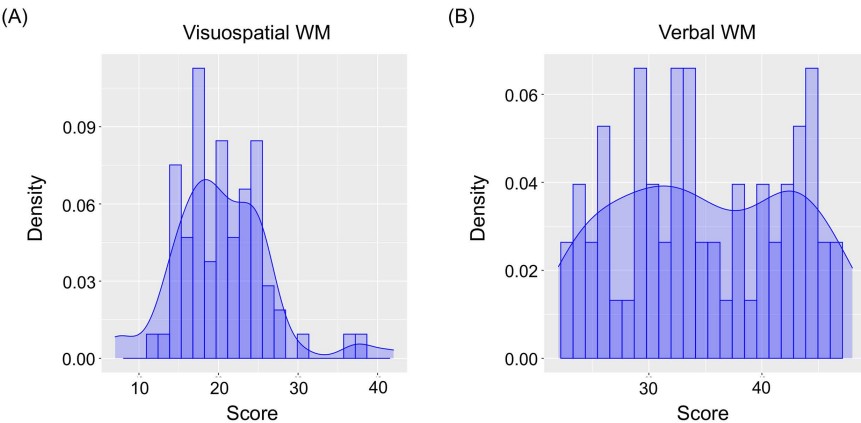

**Fig 2. Distribution of working memory (WM) assessment test scores in the development phase. (A)** Distribution of scores of comparative line span (visuospatial WM). **(B)** Distribution of scores of backward digit span (verbal WM).

## Baseline characteristics of trainees who received education

Sixty trainees underwent the WM assessment test (Fig 1) and were randomly allocated to one of the two groups: 31 in the Matched-E group and 29 in the Unmatched-E group. In the Matched-E group, 20 trainees were visuospatial WM-dominant, and 11 were verbal WM-dominant. In the Unmatched-E group, 19 trainees were visuospatial WM-dominant and 10 were verbal WM-dominant. Among the 60 trainees in the validation phase, 21 in the Matched-E group and 19 in the Unmatched-E group completed the education program and the final diagnosis test. No differences in background characteristics were observed between the two groups (Table 1). The difference in deviation between visuospatial and verbal WM varied among the trainees (Fig 3).

**Table 1. Background characteristics of the study participants in the validation phase.**

|  | Matched-E (n=21) | Unmatched-E (n=19) | P-value |
|---|---|---|---|
| Age (year, mean±SD) | 26.8±2.4 | 28.4±3.9 | 0.091 |
| Sex (male/ female) | 12/ 9 | 13/ 6 | 0.527 |
| Years of doctor experience (< 1 year/ 1–2 years) | 4/ 17 | 6/ 13 | 0.473 |
| Dominant WM (visuospatial/ verbal) | 16/ 5 | 13/ 6 | 0.727 |
| Deviation value of WM assessment test |  |  |  |
| Comparative line span (visuospatial WM) (mean±SD) | 55.3±11.3 | 52.0±8.5 | 0.224 |
| Backward digit span (verbal WM) (mean±SD) | 51.0±13.7 | 46.8±9.6 | 0.191 |

Matched-E, matched education group; Unmatched-E, unmatched education group; SD, standard deviation; WM, working memory; The values of age and deviation of WM assessment test were expressed as mean±standard deviation. P<0.05 was considered statistically significant.

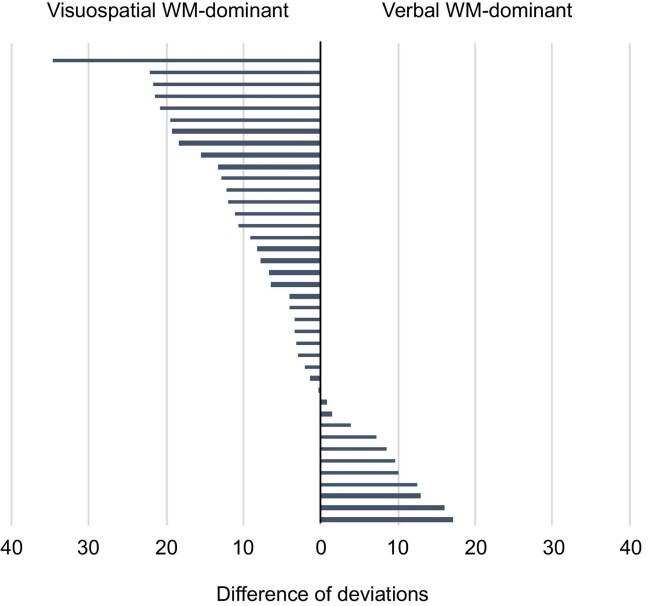

**Fig 3. Difference in deviation between visuospatial and verbal working memory (WM) of trainees in the validation phase.** The difference between the deviations of visuospatial and verbal WM was displayed as a bar graph for each trainee who completed the randomized controlled trial. The visuospatial WM is dominant if the bars are shown in the left direction, and the verbal WM is dominant if the bars are displayed in the right direction.

## Diagnostic accuracy in Matched-E and Unmatched-E groups

The diagnostic accuracy for colorectal polyps was significantly higher in the Matched-E group than that in the Unmatched-E group (61.6% vs. 53.9%, P = 0.009) (Fig 4).

## Sub-group analysis of the diagnostic accuracy

The accuracy for distinguishing non-neoplastic lesions from neoplastic lesions was significantly higher in the Matched-E group than that in Unmatched-E (68.0% vs. 48.4%, P 0.002) (Fig 5). Sub-analysis based on the lesion pathology revealed that the diagnostic accuracy for the non-neoplastic lesions was significantly higher in the Matched-E group than that in the Unmatched-E group (68.0% vs. 48.4%, P = 0.002). In contrast, no difference was observed between the two groups for diagnostic accuracy of adenoma/intramucosal cancer (56.7% vs. 56.7%, P = 0.998) and invasive cancer (57.8% vs. 61.7%, P = 0.515). Additionally, no difference was observed in the diagnostic accuracy between the Matched-E and Unmatched-E groups in terms of sex (men, 61.0% vs. 54.0%, P = 0.091; women, 62.4% vs. 53.7%, P = 0.064). Analysis based on years of doctoral experience also showed no difference between the two groups (< 1-year experience, 66.5% vs. 51.3%, P = 0.072; 1–2-year experience, 60.5% vs. 51.3%, P = 0.072). No difference was observed in diagnostic accuracy between the two groups for the pattern recognition- and NICE classification-based programs (pattern recognition, 62.9% vs. 57.3%, P = 0.145; NICE classification, 57.6% vs. 52.3%, P = 0.326). The response time was 7.14 ± 6.09 s in the Matched-E group and 6.05 ± 4.78 s in the Unmatched-E group, with no significant difference.

## Discussion

A major limitation of endoscopy education is the difficulty in determining the optimal learning method for trainees, since they learn in ways that are appropriate for them. Although one study reported the effectiveness of educational interventions tailored to the personalities of endoscopy specialists to improve the quality of colonoscopy [18], no report is available on tailored education for trainees. The WM refers to the ability to hold and briefly process information within the brain. Several studies have shown that WM is closely related to learning performance in children [19–21]; its usefulness has also

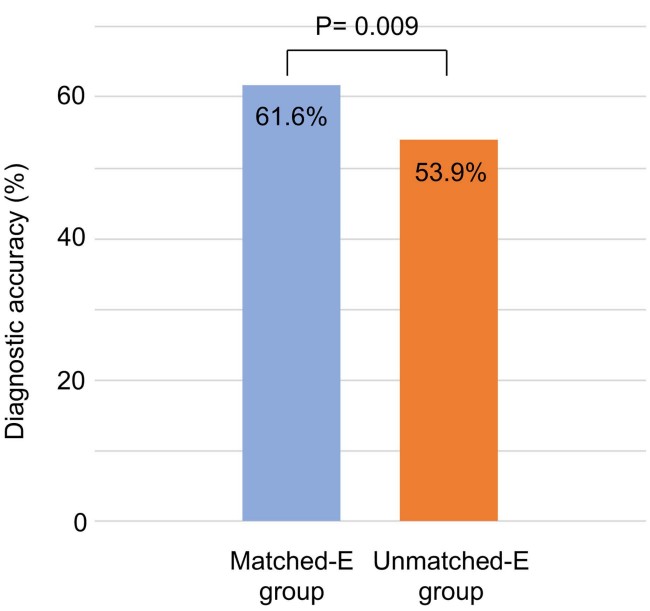

**Fig 4. Diagnostic accuracy in the Matched-E and Unmatched-E groups.** Diagnostic accuracy was significantly higher in the matched education group (Matched-E group) than that in the unmatched education group (Unmatched-E group) (61.6% vs. 53.9%, P = 0.009).

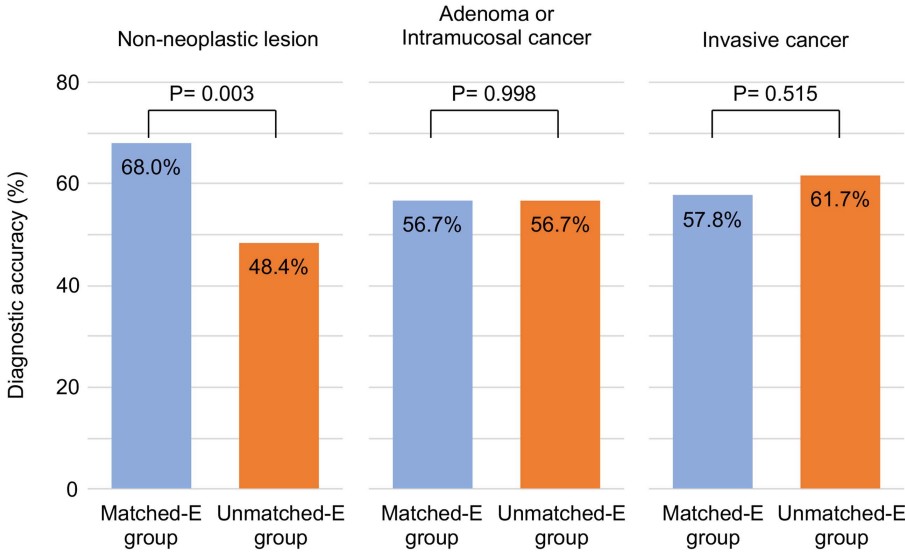

**Fig 5. Distinction accuracy for non-neoplastic lesions from neoplastic lesions.** Diagnostic accuracy for the non-neoplastic lesions was significantly higher in the matched education (Matched-E) group than that in the unmatched education (Unmatched-E) group (68.0% vs. 48.4%, P=0.002). In contrast, no difference was observed between the two groups for adenoma/intramucosal cancer (56.7% vs. 56.7%, P=0.998) and invasive cancer (57.8% vs. 61.7%, P=0.515).

been demonstrated in adults [22]. Accordingly, in this study, we developed a methodology for qualifying individual WM for endoscopy education. Specifically, this proof-of-concept study explored the feasibility of implementing tailored education based on trainees' WM profiles. The study proposed a novel concept: integrating cognitive psychology into endoscopic training. Accordingly, our results demonstrated that the Matched-E group yielded a diagnostic accuracy of 61.6%, which was significantly higher than that of the Unmatched-E group.

Although the absolute difference between the two groups in this study was 7.7%, a previous report showed a 13.6% improvement in diagnostic accuracy by a web-based image education program using the NICE classification [23]. However, in that report, the authors educated the trainees based on the NICE classification, using polyp images over 10 min. In contrast, our study explained the NICE classification in words only and did not show actual polyp images; therefore, it did not have the same educational effect as previously reported. Notably, our results showed that a verbal WM-dominant trainee was able to diagnose polyps on the first look without viewing any polyp images. However, in actual educational settings, trainees learn by simultaneously observing polyps and referring to the NICE classification, which is expected to have a greater educational effect than our results. Nonetheless, our results suggest that the efficacy of verbal-oriented classification systems, such as the NICE classification, is maximized by targeting a suitable trainee based on their learning characteristics.

However, the diagnostic accuracy in the Unmatched-E group was only 53.9% (approximately 50%), which was almost identical to the results of the intuitive answers. This was because the enrolled trainees did not have endoscopic experience and were unfamiliar with endoscopic diagnosis. This result indicates that a short 10-min teaching session has minimal effect on learning for teaching methods that are not suited to the characteristics of learning. Moreover, in actual educational settings, it is often difficult for busy supervisors to take the time to educate trainees; therefore, a method that is sufficiently effective in a short period is warranted. In this regard, the significantly higher diagnostic accuracy in the Matched-E group indicates that even complete beginners, with no experience, can benefit from 10 min of education tailored to their learning characteristics.

Discriminating non-neoplastic lesions from neoplastic ones is vital in clinical practice because it reduces unnecessary treatments. In particular, the overdiagnosis of non-neoplastic lesions as neoplastic lesions is harmful to patients and healthcare economics. Although the recent increase in the use of computer-aided detection systems has improved the polyp detection rate, it has also led to the detection of polyps that should not be removed, resulting in excessive treatment [24]. Notably, this study demonstrated a significant difference in the diagnostic accuracy for non-neoplastic polyps between the Matched-E and Unmatched-E groups. In contrast, no difference was observed between the diagnoses of adenomas and intramucosal or invasive cancers. Trainees can easily acquire the skill of distinguishing non-neoplastic and neoplastic lesions even after brief education because these lesions exhibit distinct surface and vascular patterns. In contrast, among the neoplastic lesions, adenomas and cancer share similar surface and vascular patterns, making it challenging to distinguish between them.

The lack of improvement in diagnostic accuracy for adenomas and cancers represents the most important limitation of this study for two main reasons. First, the morphological similarity of surface features between adenomas and cancers likely limited the effectiveness of short-term training. Second, because colorectal polyp diagnosis is inherently a visual task, providing a text-only training program to participants with predominantly verbal WM may have created a mismatch between instructional modality and task demands. Indeed, even within the Matched-E group, diagnostic accuracy for adenoma and intramucosal cancer was lower in the verbal-education subgroup than in the visuospatial-education subgroup (45.5% vs. 61.3%, P = 0.016; Supplementary Table 3). Thus, verbal instruction may not have been fully aligned with the ultimate goal of improving polyp diagnostic ability. In particular, the second reason suggests that direct application of this approach to clinical practice is challenging. It is not feasible in real-world training to fully segregate trainees into verbal or visual WM groups. Reports on short-term training using classification systems have incorporated image-based and text-based materials [6,7], further suggesting that purely verbal or purely visual approaches are inefficient. For inherently visual diagnostic tasks such as polyp classification, extended verbal-only training is unlikely to be effective for individuals with verbal WM dominance. A more practical strategy would be to apply hybrid methods, focusing on visual education for visual WM-dominant trainees while combining verbal and visual instruction for verbal WM-dominant trainees.

Consequently, fostering mutual understanding between trainees and trainers regarding the most suitable instructional approaches may enable the implementation of more efficient and focused training, even within the constraints of busy clinical environments. Furthermore, understanding a trainee's learning characteristics through the WM assessment may be applied beyond the acquisition of polyp diagnostic skills. For example, verbal WM-dominant trainees tend to organize and understand procedures and operating methods logically and verbally, and may be better able to extract meaning from verbal explanations than visuospatial WM-dominant trainees. Additionally, structuring and explaining each step of a procedure verbally in endoscopic submucosal dissection techniques may support efficient learning for verbal WM-dominant trainees. The results of this study also indicate that a pattern recognition-based training program is effective for teaching visuospatial WM-dominant trainees. Considering this result, pattern recognition may also be effective in learning other skills, such as endoscopic resection and endoscopic retrograde cholangiopancreatography (ERCP). For instance, the effectiveness of repeated motion training using simulation models has been demonstrated in ERCP education [25]. Thus, visuospatial-WM-dominant trainees may be suitable for learning via patterned educational methods. Additionally, the rapid insertion of a colonoscope requires spatial comprehension to understand the shape of the intestinal lumen in three dimensions. Moreover, WM can change with training [26]. Thus, it may be possible to devise a method to learn ERCP skills and colonoscope insertion techniques by modifying the WM itself through training, despite a low visuospatial WM score.

Nonetheless, this study has some limitations. First, only two WM assessment test tasks were used to determine the WM profiles of the trainees. In the future, the WM assessment of trainees can be made more accurate by imposing tasks similar to the original HUCRoW. Second, the possibility that fraud occurred cannot be ruled out because the trainees did not monitor where they received education or polyp diagnostic testing. Therefore, these results may have been underestimated or overestimated. Third, this study included trainees affiliated with university hospitals in Japan. Additionally,

their baseline skill levels may differ from those at other institutions in Japan or abroad; therefore, the results may not be generalizable to other institutions. Fourth, the analysis by sex or years of experience may not have detected differences in distinction accuracy between the two groups because of the reduced number of participants in the subgroup analysis. Fifth, a substantial number of dropouts were observed in this study. The enrolled participants were hospital residents from various departments, including those outside of the gastroenterology department. Therefore, not all participants were entirely motivated and committed to the study, despite their initial agreement to participate. Nevertheless, the final number of participants analyzed reached the same sample size as the preceding one after excluding these dropouts. Furthermore, based on the actual observed data (delta = 7.7%, standard deviation = 9.6%), the number of analyzed participants yielded adequate power (94.3%). This reinforces the validity of this study. Finally, the long-term effects of the pedagogy used in this study are undetermined. Consequently, a long-term prospective study is warranted to confirm the results of this study.

In conclusion, trainees with no endoscopic experience can efficiently acquire endoscopic diagnostic skills by receiving education suited to their WM profile based on cognitive and psychological characteristics. However, the lack of improvement in diagnostic accuracy for neoplastic lesions after short-term training remains a notable limitation. In clinical practice, it is difficult to fully stratify trainees into verbal or visual WM groups and implement strictly modality-specific training. Addressing this limitation will be essential to achieve truly individualized education.

## Supporting information

**S1 Fig. Comparative line span.** (A) First, respondents memorized the line length shown on the left. Subsequently, they answered whether the line on the right that appears is the same length as the line they memorized. (B) They pointed to the line's location memorized in (A) in the 3 × 3 square where the line is not drawn. These (A) and (B) were first presented one question at a time, then two, then three, and so on, increasing the number and location of lines to be memorized. (PDF)

**S2 Fig. Backward digit span.** (A) Respondents heard numbers from 1 to 9 through audio and memorized them. (B) They then selected the numbers from a list of candidates in reverse order. The numbers were presented in random order, as shown in the figure. Each time respondents answered a question correctly, the number of numbers presented increased. (PDF)

**S3 Fig. Difference in deviation between visuospatial and verbal working memory (WM) of endoscopists in the development phase.** The difference between the deviations of the visuospatial and verbal WM was displayed as a bar graph for each endoscopist participating in the development phase. The visuospatial WM is dominant if the bars are shown in the left direction, and the verbal WM is dominant if the bars are displayed in the right direction. (PDF)

**S1 Table. NICE classification.** (DOCX)

**S2 Table. Baseline characteristics of endoscopists for the development of standard working memory profile.** (DOCX)

**S3 Table. Diagnostic accuracy according to the training program in the Matched-E and Unmatched-E groups.** (DOCX)

## Acknowledgments

We thank Dr. Kiichi Sato for supporting the recruitment of endoscopists during the developmental phase.

## Author contributions

**Conceptualization:** Kentaro Mochida, Fumiaki Ishibashi, Masamichi Yuzawa, Sho Suzuki.

**Data curation:** Kentaro Mochida, Fumiaki Ishibashi, Daisuke Suto, Toshihiro Nishizawa, Yasunari Sakamoto, Mikinori Kataoka, Mitsuru Esaki, Chika Kusano, Takuji Gotoda, Kenichi Konda, Jun Arimoto, Ryu Tanaka, Kohei Yamanouchi, Masao Okubo, Kazuma Fujimoto, Tomohiro Kawakami, Mizuki Nagai.

**Formal analysis:** Kentaro Mochida, Fumiaki Ishibashi.

**Funding acquisition:** Fumiaki Ishibashi.

**Investigation:** Kentaro Mochida, Fumiaki Ishibashi.

**Methodology:** Fumiaki Ishibashi, Masamichi Yuzawa.

**Project administration:** Fumiaki Ishibashi.

**Resources:** Fumiaki Ishibashi, Masamichi Yuzawa.

**Software:** Fumiaki Ishibashi, Masamichi Yuzawa.

**Supervision:** Sho Suzuki.

**Validation:** Sho Suzuki.

**Visualization:** Kentaro Mochida, Fumiaki Ishibashi.

**Writing – original draft:** Kentaro Mochida, Fumiaki Ishibashi.

**Writing – review & editing:** Sho Suzuki.

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
