## [Decision Letter · Decision Letter 0]

17 Jul 2025

PONE-D-24-56097Personalized education approach based on cognitive psychology for endoscopic diagnosis: a multicenter randomized trialPLOS ONE

Dear Dr. Ishibashi,

Thank you for submitting your manuscript to PLOS ONE. After careful consideration, we feel that it has merit but does not fully meet PLOS ONE’s publication criteria as it currently stands. Therefore, we invite you to submit a revised version of the manuscript that addresses the points raised during the review process.

We look forward to receiving your revised manuscript.

Kind regards,

Naoki Asano, M.D., Ph.D.

Academic Editor

PLOS ONE

Journal Requirements:

“This work was supported by International University of Health and Welfare Research Funds for Academic Year 2023.”

Please state what role the funders took in the study.  If the funders had no role, please state: "The funders had no role in study design, data collection and analysis, decision to publish, or preparation of the manuscript.” If this statement is not correct you must amend it as needed. 

3. Please note that funding information should not appear in the Acknowledgments section or other areas of your manuscript. We will only publish funding information present in the Funding Statement section of the online submission form. Please remove any funding-related text from the manuscript. 

“FI received honoraria for lectures from the Fujifilm Corporation. ME is a consultant for AI Medical Service, Inc. CK received honoraria for lectures from the Fujifilm Corporation and Olympus Corporation. TG received honoraria for lectures from the Fujifilm Corporation, Fujifilm Medical, and Olympus Corporation. SS received honoraria for lectures from the Fujifilm Corporation and Olympus Corporation. The authors declare no conflicts of interest.”

**Additional Editor Comments:**

Please refer to the detailed review below in this email.

Both reviewers raised concerns regarding the diagnostic accuracy for neoplastic lesions, and these issues should be thoroughly discussed.

Reviewers' comments:

Reviewer's Responses to Questions

**Comments to the Author**

1. Is the manuscript technically sound, and do the data support the conclusions?

Reviewer #1: Yes

Reviewer #2: Partly

2. Has the statistical analysis been performed appropriately and rigorously? 

Reviewer #1: Yes

Reviewer #2: Yes

3. Have the authors made all data underlying the findings in their manuscript fully available?

Reviewer #1: Yes

Reviewer #2: Yes

4. Is the manuscript presented in an intelligible fashion and written in standard English?

Reviewer #1: Yes

Reviewer #2: Yes

5. Review Comments to the Author

Reviewer #1: I would like to thank for the opportunity to review the manuscript entitled "Personalized Education Approach Based on Cognitive Psychology for Endoscopic Diagnosis: A Multicenter Randomized Trial."

The strengths of this manuscript are that it introduces a new perspective of utilizing working memory (WM) in endoscopic diagnosis education.

It aims to improve diagnostic accuracy by considering the differences between visuospatial WM and verbal WM and applying the most suitable educational method for each learner.

It was designed as a multicenter randomized controlled trial (RCT) across four facilities, so there is a certain degree of generalizability.

There are a few points to confirm.

The sample size was 60 trainees at the start of the study, but only 40 were ultimately analyzed, so there is a possibility that the statistical power is insufficient.

I am concerned about the large proportion of "dropouts" in the RCT and the fact that the reasons for dropouts were not explained in detail.

The improvement in diagnostic accuracy for non-neoplastic lesions was significant (68.0% vs. 48.4%, P=0.002), and I agree that it may contribute to reducing unnecessary treatment.

On the other hand, there was no difference in diagnostic accuracy for neoplastic lesions (56.7% vs. 56.7%, P=0.998). What is the reason for this?

There has been only limited discussion about the applicability of endoscopic education using WM profiling to actual clinical practice, so please add more discussion.

Although a brief 10-minute education session has been shown to be effective, the long-term effects (e.g., maintaining diagnostic accuracy several months later) have not been evaluated.

Reviewer #2: Dear authors,

Thank you for submitting your manuscript, titled Personalised Education Approach Based on Cognitive Psychology for Endoscopic Diagnosis: A Multicentre Randomised Trial (Manuscript Number: PONE-D-24-56097).

I found your approach to personalising endoscopic training based on working memory profiles to be both interesting and conceptually strong. The design of your multicentre randomised trial is also well executed.

However, after careful review, I have significant concerns about the practical implications of your findings and the generalisability of your proposed methodology, which I believe must be thoroughly addressed.

My main points are:

Clinical relevance for neoplastic lesions: Crucially, your study found no significant difference in diagnostic accuracy for neoplastic lesions (adenoma/cancer) between the two educational groups. As an accurate diagnosis of these lesions is the primary goal of endoscopy, the lack of improvement in this area significantly limits the clinical utility of your personalised educational approach. While differences were observed for non-neoplastic lesions, their overall accuracy remained relatively low (68.0%).

Mismatch of Learning Approach with Task Nature: Endoscopic diagnosis, particularly when using systems such as the NICE classification, is a highly visual task that requires pattern recognition. Using a 'descriptive text'-based education approach for verbally dominant learners, while aligning with their working memory (WM) type, may not be the most effective way to teach a task that relies so heavily on visual cues. It is difficult to extrapolate how this methodology would generally enhance visual diagnostic abilities.

While your initial concept is commendable, the fact that your intervention did not significantly improve accuracy for the most clinically important type of lesion, combined with the potential mismatch between the learning method and the visual nature of endoscopic diagnosis, raises serious questions about the effectiveness and general applicability of this personalised education strategy.

Unless these critical issues are addressed and a clear pathway is presented on how this methodology or similar approaches can genuinely enhance future endoscopic learning, based on your current results, I cannot recommend acceptance of the manuscript.

Thank you.

6. PLOS authors have the option to publish the peer review history of their article (what does this mean? ). If published, this will include your full peer review and any attached files.

**Do you want your identity to be public for this peer review?** For information about this choice, including consent withdrawal, please see our Privacy Policy .

Reviewer #1: No

Reviewer #2: No

---

## [Author Response · Author response to Decision Letter 1]

12 Aug 2025

August 7, 2025

Emily Chenette

Editor-in-Chief

PLOS ONE

Dear Editor:

On behalf of all the authors, I thank you for reviewing our manuscript titled “Personalized education approach based on cognitive psychology for endoscopic diagnosis: a multicenter randomized trial.”

We appreciate your valuable comments that have helped us to improve our manuscript substantially. We have revised the manuscript in line with your suggestions and have highlighted all related changes in red. Additionally, we have provided point-by-point responses to the reviewers’ comments attached below.

Thank you for your consideration. I look forward to hearing from you.

Sincerely,

Fumiaki Ishibashi

Department of Gastroenterology, International University of Health and Welfare Ichikawa Hospital

6-1-14, Konodai, Ichikawa-shi, Chiba 272-0827, Japan

Email: f.jazzmaster@gmail.com

Phone: +81-47-375-1111

Fax: +81-47-373-4921

Editorial and Journal Office Requirements

Comment 1:

Response to Comment 1:

We have revised our manuscript according to the PLOS ONE formatting style. File names have also been updated accordingly.

Comment 2:

Thank you for stating the following financial disclosure:

“This work was supported by International University of Health and Welfare Research Funds for Academic Year 2023.”

Please state what role the funders took in the study. If the funders had no role, please state: "The funders had no role in study design, data collection and analysis, decision to publish, or preparation of the manuscript.” If this statement is not correct you must amend it as needed.

Response to Comment 2:

As the funders had no role in the study, we have added the following statement in the cover letter as requested: "The funders had no role in study design, data collection and analysis, decision to publish, or preparation of the manuscript."

Comment 3:

Funding information should not appear in the Acknowledgments section or elsewhere in the manuscript. We will only publish funding information present in the Funding Statement section of the online submission form. Please remove any funding-related text from the manuscript.

Response to Comment 3:

As per the editorial office’s indication, funding information has been removed from the Acknowledgments and other sections of the manuscript.

Comment 4:

Thank you for stating the following in the Competing Interests section:

“FI received honoraria for lectures from the Fujifilm Corporation. ME is a consultant for AI Medical Service, Inc. CK received honoraria for lectures from the Fujifilm Corporation and Olympus Corporation. TG received honoraria for lectures from the Fujifilm Corporation, Fujifilm Medical, and Olympus Corporation. SS received honoraria for lectures from the Fujifilm Corporation and Olympus Corporation. The authors declare no conflicts of interest.”

Response to Comment 4:

We have revised the Competing Interests section in the cover letter to include the following statement as per your suggestion: "This does not alter our adherence to PLOS ONE policies on sharing data and materials."

Comment 5:

We note that you have included the phrase “data not shown” in your manuscript. Unfortunately, this does not meet our data sharing requirements. PLOS does not permit references to inaccessible data. We require that authors provide all relevant data within the paper, Supporting Information files, or in an acceptable, public repository. Please add a citation to support this phrase or upload the data that corresponds with these findings to a stable repository (such as Figshare or Dryad) and provide and URLs, DOIs, or accession numbers that may be used to access these data. Or, if the data are not a core part of the research being presented in your study, we ask that you remove the phrase that refers to these data.

Response to Comment 5:

As the data were not a core part of the research being presented in this study, the related sentences were removed from the manuscript.

Comment 6:

Your ethics statement should only appear in the Methods section of your manuscript. If your ethics statement is written in any section besides the Methods, please move it to the Methods section and delete it from any other section. Please ensure that your ethics statement is included in your manuscript, as the ethics statement entered into the online submission form will not be published alongside your manuscript.

Response to Comment 6:

We have confirmed that the ethics statement appears only in the Methods section in the revised manuscript.

Reviewer #1

Comment 1:

The sample size was 60 trainees at the start of the study, but only 40 were ultimately analyzed, so there is a possibility that the statistical power is insufficient.

Response to Comment 1:

Thank you for your pertinent comment. As you have pointed out, the final analysis included 40 participants, which means 20 participants dropped out from the initial 60 registered at the time of enrollment. However, per our analysis, the minimum sample size required to assess the study endpoint was 34 participants. Therefore, 40 participants was deemed to be sufficient to determine the endpoint with a 10% difference in diagnostic accuracy (α = 0.05, power = 0.80). Moreover, based on the actual observed data (delta= 7.7%, standard deviation= 9.6%), the number of analyzed participants yielded adequate statistic power (94.3%). We have explained this in the Discussion section in the revised manuscript.

Page 21 Line 8–10 (Discussion): Furthermore, based on the actual observed data (delta= 7.7%, standard deviation= 9.6%), the number of analyzed participants yielded adequate statistic power (94.3%). This reinforces the validity of this study.

Comment 2: I am concerned about the large proportion of "dropouts" in the RCT and the fact that the reasons for dropouts were not explained in detail.

Response to Comment 2:

Thank you for your pertinent comment. The participants in this study were hospital residents who belonged to various departments, including those other than gastroenterology. Therefore, not all participants were fully motivated and committed to the study, despite their initial agreement to participate. Nevertheless, after excluding these dropouts, the final number of participants analyzed reached the same sample size as the preceding one. The detailed reasons for dropout have now been explained in the figure legend to maintain the transparency of the study flow (Fig 1. Study flow diagram). Additionally, an explanation has been included in the Discussion section.

Page 21, Line 3–8 (Discussion): Fifth, a substantial number of dropouts were observed in this study. The enrolled participants were hospital residents from various departments, including those outside of the gastroenterology department. Therefore, not all participants were entirely motivated and committed to the study, despite their initial agreement to participate. Nevertheless, the final number of participants analyzed reached the same sample size as the preceding one after excluding these dropouts.

Comment 3: The improvement in diagnostic accuracy for non-neoplastic lesions was significant (68.0% vs. 48.4%, P=0.002), and I agree that it may contribute to reducing unnecessary treatment.

On the other hand, there was no difference in diagnostic accuracy for neoplastic lesions (56.7% vs. 56.7%, P=0.998). What is the reason for this?

Response to Comment #3:

Thank you for your pertinent comment. The primary reason for the lack of difference in the diagnostic accuracy for neoplastic lesions is that non-neoplastic and neoplastic lesions exhibit distinct surface and vascular patterns, allowing trainees to easily acquire the skill of distinguishing these lesions even after brief education. In contrast, neoplastic lesions such as adenoma and cancer share similar surface and vascular patterns, making it challenging to distinguish between them. Therefore, a longer training intervention may be necessary to observe the difference in distinction accuracy for neoplastic lesions between the Matched-E and Unmatched-E groups. This explanation has been included in the revised Discussion section.

Page 18 Line 17– Page 19 Line 4 (Discussion): Trainees can easily acquire the skill of distinguishing non-neoplastic and neoplastic lesions even after brief education because these lesions exhibit distinct surface and vascular patterns. In contrast, among the neoplastic lesions, adenomas and cancer share similar surface and vascular patterns, making it challenging to distinguish between them. Thus, a longer training intervention may be necessary to observe the difference in distinction accuracy for neoplastic lesions between the Matched-E and Unmatched-E groups.

Comment 4: There has been only limited discussion about the applicability of endoscopic education using WM profiling to actual clinical practice, so please add more discussion.

Response to Comment 4:

We appreciate your valuable comment. Notably, our proof-of-concept study demonstrated the potential for tailoring endoscopic education according to individual WM profiles, particularly in the context of acquiring diagnostic skills for colorectal polyps through a brief, 10-min training session. The results revealed a statistically significant benefit in short-term educational effectiveness when trainees were provided with instructional methods aligned with their WM profile, broadly categorized as either verbal- or visuospatial-dominant. However, endoscopic training in real-world clinical settings is usually a combination of verbal and visuospatial modalities. Therefore, mutual understanding between trainees and trainers regarding the most suitable instructional approaches may enable the implementation of more efficient and focused training, even within the constraints of busy clinical environments. This discussion about the applicability of endoscopic education using WM profiling to actual clinical practice have been included in the Discussion section. Additionally, we have explained the possibility of clinical application of WM-based training beyond the colorectal polyp diagnosis, eg, ESD and ERCP in the following sentences.

Page 19 Line 5–11 (Discussion): However, in real-world clinical settings, endoscopic training is rarely limited to purely verbal or purely visuospatial modalities; rather, a combination of both is typically employed. Therefore, employing either description text-based learning or pattern recognition learning alone as a standalone educational method, as used in this study, would be of limited practical value. Consequently, fostering mutual understanding between trainees and trainers regarding the most suitable instructional approaches may enable the implementation of more efficient and focused training, even within the constraints of busy clinical environments.

Comment 5: Although a brief 10-minute education session has been shown to be effective, the long-term effects (e.g., maintaining diagnostic accuracy several months later) have not been evaluated.

Response to Comment 5:

Thank you for your insightful comment. In real-world educational settings, training rarely terminates with a single session; rather, it is a repetitive and ongoing process. Therefore, assessing whether a one-time educational intervention has a measurable impact several months later may be of clinical relevance. Nevertheless, we acknowledge this as a limitation of our study and have added the following sentences to the Discussion section.

Page 21 Line 10-12 (Discussion): Finally, the long-term effects of the pedagogy used in this study are undetermined. Consequently, a long-term prospective study is warranted to confirm the results of this study.

Reviewer #2

Comment 1:

Clinical relevance for neoplastic lesions: Crucially, your study found no significant difference in diagnostic accuracy for neoplastic lesions (adenoma/cancer) between the two educational groups. As an accurate diagnosis of these lesions is the primary goal of endoscopy, the lack of improvement in this area significantly limits the clinical utility of your personalised educational approach. While differences were observed for non-neoplastic lesions, their overall accuracy remained relatively low (68.0%).

Response to Comment 1:

Thank you for your pertinent comment. The primary endpoint of this study was the proportion of correct diagnoses for non-neoplastic lesions, adenomas/intramucosal cancers, and invasive cancers, representing the diagnostic accuracy for each pathological category. As a secondary endpoint, we evaluated the ability to distinguish each pathological category from the other two. The results showed that the ability to differentiate non-neoplastic lesions from the other two categories (adenomas/intramucosal cancers and invasive cancers) was significantly higher in the Matched-E group than that in the Unmatched-E group. This finding indicates the acquisition of clinically important diagnostic competence in distinguishing non-neoplastic from neoplastic lesions. We have added the following sentences in the Method section to clearly explain this aim. Additionally, we have discussed the differential acquisition of diagnostic skills between neoplastic and non-neoplastic lesions in the Discussion section.

Moreover, as this study was positioned as a proof-of-concept study to explore the feasibility of implementing tailored education based on trainees’ WM profiles, its primary significance lies in proposing a novel concept: the integration of cognitive psychology into endoscopic training. As an initial step toward proposing this concept, we evaluated outcomes that could be measured within a short period, using the minimum required sample size for a clinical study. Therefore, we believe that the lack of statistically significant improvement in the differential diagnosis of neoplastic lesions, and the relatively low diagnostic accuracy of 68.0% for non-neoplastic lesions should not be viewed as critical limitations. We have included a discussion of the study's proof-of-concept nature in the revised manuscript.

Page 11 Line 5–6 (Methods): Additionally, the diagnostic accuracy according to the lesion pathology was evaluated to determine the ability to distinguish each pathological category from the other two.

Page 16 Line 17– Page 17 Line 1 (Discussion): Specifically, this proof-of-concept study explored the feasibility of implementing tailored education based on trainees’ WM profiles. The study proposed a novel concept: integrating cognitive psychology into endoscopic training.

Page 18 Line 17– Page 19 Line 4 (Discussion): Trainees can easily acquire the skill of distinguishing non-neoplastic and neoplastic lesions even after brief education because these lesions exhibit distinct surface and vascular patterns. In contrast, among the neoplastic lesions, adenomas and cancer share similar surface and vascular patterns, making it challenging to distinguish between them. Thus, a longer training intervention may be necessary to observe the difference in distinction accuracy for neoplastic lesions between the Matched-E and Unmatched-E groups.

Comment 2:

Mismatch of Learning Approach with Task Nature: Endoscopic diagnosis, particularly when using systems such as the NICE classification, is a highly visual task that requires pattern r

---

## [Decision Letter · Decision Letter 1]

27 Aug 2025

PONE-D-24-56097R1Personalized education approach based on cognitive psychology for endoscopic diagnosis: a multicenter randomized trialPLOS ONE

Dear Dr. Ishibashi,

Thank you for submitting your manuscript to PLOS ONE. After careful consideration, we feel that it has merit but does not fully meet PLOS ONE’s publication criteria as it currently stands. Therefore, we invite you to submit a revised version of the manuscript that addresses the points raised during the review process.

**ACADEMIC EDITOR: **Thank you for submitting your revised manuscript to PLOS One.

Although significancant changes have been made in response to the reviewer's comments and have conributed to improving the quality of the manuscript, some of the reviewer's concerns still remain insufficiently addressed. 

Specifically, please consider addressing the following points: recognition of the limitation due to the absence of improvement in the diagnosis of neoplastic lesions, indicating possible underlying mechanism for the lack of improvement in the diagnosis of neoplastic lesions and strategies to overcome this limitation in the future.

We look forward to receiving your revised manuscript.

Kind regards,

Naoki Asano, M.D., Ph.D.

Academic Editor

PLOS ONE

Journal Requirements:

Reviewers' comments:

Reviewer's Responses to Questions

**Comments to the Author**

1. If the authors have adequately addressed your comments raised in a previous round of review and you feel that this manuscript is now acceptable for publication, you may indicate that here to bypass the “Comments to the Author” section, enter your conflict of interest statement in the “Confidential to Editor” section, and submit your "Accept" recommendation.

Reviewer #1: All comments have been addressed

Reviewer #2: (No Response)

2. Is the manuscript technically sound, and do the data support the conclusions?

Reviewer #1: Yes

Reviewer #2: Yes

3. Has the statistical analysis been performed appropriately and rigorously? 

Reviewer #1: Yes

Reviewer #2: Yes

4. Have the authors made all data underlying the findings in their manuscript fully available?

Reviewer #1: Yes

Reviewer #2: Yes

5. Is the manuscript presented in an intelligible fashion and written in standard English?

Reviewer #1: Yes

Reviewer #2: Yes

6. Review Comments to the Author

Reviewer #1: The revisions have been made appropriately in response to the comments, and I have no additional remarks. I consider this manuscript suitable for acceptance.

Reviewer #2: Further comment on Author Response

Thank you for the revision. While the clarification of the study’s proof-of-concept nature and the additional discussion regarding non-neoplastic lesions are acknowledged, the response does not sufficiently address the principal concern regarding the lack of improvement in neoplastic lesion diagnosis.

Recognition of a Major Limitation

The absence of improvement in the diagnosis of neoplastic lesions—particularly adenomas and cancers—should be explicitly recognized as a major limitation, even in the context of a proof-of-concept study. Minimizing the importance of this finding risks underrepresenting its impact on the clinical applicability of the proposed educational approach.

Plausible Underlying Mechanism

In addition to morphological similarity between adenomas and early cancers, a plausible contributing factor may be a mismatch between the instructional modality and the inherently visual nature of the diagnostic task. A text-based, verbal WM–aligned approach may be insufficient for developing advanced visual pattern recognition skills. This possible mechanism should be incorporated into the Discussion to provide a more comprehensive interpretation of the findings.

Specification of Future Directions

The statement that “longer training may be necessary” is insufficiently specific. The Discussion should propose concrete strategies for overcoming the identified limitation, such as incorporating visually intensive instructional components or hybrid methods that integrate both visual and verbal modalities, tailored to WM profiles for tasks with high visual demands.

Overall, while the concept of WM-tailored education is novel and potentially valuable, its future clinical relevance will depend on demonstrating benefits for neoplastic lesion diagnosis. The revised manuscript should therefore clearly (1) acknowledge this limitation, (2) link it to a plausible cognitive–task mismatch, and (3) outline specific, actionable strategies for future research and educational design.

7. PLOS authors have the option to publish the peer review history of their article (what does this mean? ). If published, this will include your full peer review and any attached files.

**Do you want your identity to be public for this peer review?** For information about this choice, including consent withdrawal, please see our Privacy Policy .

Reviewer #1: **Yes: ** Yoriaki Komeda

Reviewer #2: No

---

## [Author Response · Author response to Decision Letter 2]

1 Sep 2025

September 2, 2025

Emily Chenette

Editor-in-Chief

PLOS ONE

Dear Editor:

On behalf of all the authors, I would like to thank you for reviewing our manuscript titled “Personalized education approach based on cognitive psychology for endoscopic diagnosis: a multicenter randomized trial.”

We greatly appreciate your valuable comments, which have helped us substantially improve the manuscript. We have revised it according to your suggestions, with all related changes indicated in red. Additionally, we have provided detailed point-by-point responses to the reviewers’ comments, as attached below.

Thank you for your consideration. I look forward to hearing from you.

Sincerely,

Fumiaki Ishibashi

Department of Gastroenterology, International University of Health and Welfare Ichikawa Hospital

6-1-14, Konodai, Ichikawa-shi, Chiba 272-0827, Japan

Email: f.jazzmaster@gmail.com

Phone: +81-47-375-1111

Fax: +81-47-373-4921

Reviewer #2

Comment 1:

Recognition of a Major Limitation

The absence of improvement in the diagnosis of neoplastic lesions—particularly adenomas and cancers—should be explicitly recognized as a major limitation, even in the context of a proof-of-concept study. Minimizing the importance of this finding risks underrepresenting its impact on the clinical applicability of the proposed educational approach.

Response to Comment 1:

Thank you very much for this important comment. As the reviewer correctly pointed out, in our study, WM-based individualized education improved the ability to distinguish non-neoplastic from neoplastic lesions; however, it did not enhance diagnostic accuracy in differentiating adenomas from invasive cancers. We recognize this as a major limitation and agree that the educational approach used in this study cannot yet be directly applied to clinical practice.

In the revised manuscript, we have added a new paragraph to the Discussion section to explain these points, and we have modified the Conclusion to describe the implications of our findings in a more cautious manner.

Page 19, Lines 321-322 (Discussion): The lack of improvement in diagnostic accuracy for adenomas and cancers represents the most important limitation of this study for two main reasons.

Page 22, Lines 381-385 (Conclusion): However, the lack of improvement in diagnostic accuracy for neoplastic lesions after short-term training remains a notable limitation. In clinical practice, it is difficult to fully stratify trainees into verbal or visual WM groups and implement strictly modality-specific training. Addressing this limitation will be essential to achieve truly individualized education.

Comment 2:

Plausible Underlying Mechanism

In addition to morphological similarity between adenomas and early cancers, a plausible contributing factor may be a mismatch between the instructional modality and the inherently visual nature of the diagnostic task. A text-based, verbal WM–aligned approach may be insufficient for developing advanced visual pattern recognition skills. This possible mechanism should be incorporated into the Discussion to provide a more comprehensive interpretation of the findings.

Response to Comment 2:

We sincerely appreciate this important comment. We believe there are two main reasons why diagnostic accuracy for adenomas and cancers did not improve.

First, as noted in our previous revision, adenomas and cancers share morphological features such as surface structure, which makes it difficult to achieve meaningful learning effects through short-term training.

Second, as the reviewer rightly pointed out, the diagnosis of colorectal polyps inherently requires visual assessment. Therefore, when a text-based educational program was provided to participants with predominantly verbal WM, a mismatch may have occurred between the instructional modality and the visual nature of the diagnostic task. Indeed, even within the Matched-E group, diagnostic accuracy for adenomas and intramucosal cancers was lower in the verbal-education subgroup compared with the visual-education subgroup (45.5% vs. 61.3%, P= 0.016; Supplementary Table 3). This suggests that verbal instruction alone may not have been an appropriate method for fostering polyp diagnostic ability in this context. In particular, the second reason highlights why direct application of this educational approach to clinical practice is challenging. In real-world settings, it is not feasible to clearly stratify trainees into verbal- or visual-WM dominant groups and provide completely segregated training methods.

We have now explicitly acknowledged in the Discussion section that this constitutes the major limitation of our study. In addition, we have clarified that in practical educational settings, even for trainees with predominantly verbal WM, an approach relying exclusively on verbal instruction would be unrealistic.

Page 19, Lines 322-332 (Discussion): First, the morphological similarity of surface features between adenomas and cancers likely limited the effectiveness of short-term training. Second, because colorectal polyp diagnosis is inherently a visual task, providing a text-only training program to participants with predominantly verbal WM may have created a mismatch between instructional modality and task demands. Indeed, even within the Matched-E group, diagnostic accuracy for adenoma and intramucosal cancer was lower in the verbal-education subgroup than in the visuospatial-education subgroup (45.5% vs. 61.3%, P= 0.016; Supplementary Table 3). Thus, verbal instruction may not have been fully aligned with the ultimate goal of improving polyp diagnostic ability. In particular, the second reason suggests that direct application of this approach to clinical practice is challenging. It is not feasible in real-world training to fully segregate trainees into verbal or visual WM groups.

Comment 3:

Specification of Future Directions

The statement that “longer training may be necessary” is insufficiently specific. The Discussion should propose concrete strategies for overcoming the identified limitation, such as incorporating visually intensive instructional components or hybrid methods that integrate both visual and verbal modalities, tailored to WM profiles for tasks with high visual demands.

Response to Comment 3:

We are grateful for the clear guidance provided to improve the quality of our manuscript. As the reviewer correctly pointed out, simply extending the duration of training is insufficient. As noted in our responses to Comments 1 and 2, for tasks that fundamentally require the acquisition of visual diagnostic skills, providing prolonged verbal-only instruction to trainees with predominantly verbal WM is inefficient. For tasks such as polyp diagnosis, it may be appropriate to emphasize visual instruction for trainees with predominantly visual WM, whereas for those with predominantly verbal WM, a hybrid approach that combines visual training with verbal elements would be more practical.

Previous reports on short-term training for polyp differentiation using classification systems (Raghavendra M et al., GIE 2010; Ignjatovic A et al., GIE 2011) also employed visual and text-based teaching materials. These findings suggest that purely verbal or visual instruction is not efficient in practice. Thus, it would be more realistic for trainees and trainers to consider the predominant WM profile and apply a mixed, hybrid approach—placing greater emphasis on one modality but still incorporating the other—rather than relying exclusively on a single type of instruction.

In line with this suggestion, we have removed the statement “Thus, a longer training intervention may be necessary to observe the difference in distinction accuracy for neoplastic lesions between the Matched-E and Unmatched-E groups.”

Immediately following this section, we have added a new paragraph that discusses the possible reasons why diagnostic accuracy for neoplastic lesions such as adenomas and invasive cancers did not improve, and proposes more practical educational strategies to address this limitation.

Page 19, Line 332 to Page 20, Line 338 (Discussion): Reports on short-term training using classification systems have incorporated image-based and text-based materials [6,7], further suggesting that purely verbal or purely visual approaches are inefficient. For inherently visual diagnostic tasks such as polyp classification, extended verbal-only training is unlikely to be effective for individuals with verbal WM dominance. A more practical strategy would be to apply hybrid methods, focusing on visual education for visual WM-dominant trainees while combining verbal and visual instruction for verbal WM-dominant trainees.

---

## [Decision Letter · Decision Letter 2]

3 Sep 2025

Personalized education approach based on cognitive psychology for endoscopic diagnosis: a multicenter randomized trial

PONE-D-24-56097R2

Dear Dr. Ishibashi,

We’re pleased to inform you that your manuscript has been judged scientifically suitable for publication and will be formally accepted for publication once it meets all outstanding technical requirements.

Kind regards,

Naoki Asano, M.D., Ph.D.

Academic Editor

PLOS ONE

Additional Editor Comments (optional):

Reviewer #2:

Reviewers' comments:

Reviewer's Responses to Questions

**Comments to the Author**

1. If the authors have adequately addressed your comments raised in a previous round of review and you feel that this manuscript is now acceptable for publication, you may indicate that here to bypass the “Comments to the Author” section, enter your conflict of interest statement in the “Confidential to Editor” section, and submit your "Accept" recommendation.

Reviewer #2: All comments have been addressed

2. Is the manuscript technically sound, and do the data support the conclusions?

Reviewer #2: Yes

3. Has the statistical analysis been performed appropriately and rigorously? 

Reviewer #2: Yes

4. Have the authors made all data underlying the findings in their manuscript fully available?

Reviewer #2: Yes

5. Is the manuscript presented in an intelligible fashion and written in standard English?

Reviewer #2: Yes

6. Review Comments to the Author

Reviewer #2: The authors have substantially revised the manuscript in line with the reviewers’ comments. Importantly, they now explicitly acknowledge the lack of improvement in neoplastic lesion diagnosis as a major limitation, provide a plausible explanation regarding the potential mismatch between instructional modality and the visual nature of the task, and propose concrete future directions such as hybrid educational strategies.

These revisions ensure that the manuscript presents a balanced discussion of what can and cannot be concluded from the present study, while also outlining a realistic path forward for future research. In my view, the paper now succeeds in both proposing a novel concept for individualized endoscopic education and situating it within its appropriate limitations.

7. PLOS authors have the option to publish the peer review history of their article (what does this mean? ). If published, this will include your full peer review and any attached files.

**Do you want your identity to be public for this peer review?** For information about this choice, including consent withdrawal, please see our Privacy Policy .

Reviewer #2: No

---

## [Editor Report · Acceptance letter]

PONE-D-24-56097R2

PLOS ONE

Dear Dr. Ishibashi,

I'm pleased to inform you that your manuscript has been deemed suitable for publication in PLOS ONE. Congratulations! Your manuscript is now being handed over to our production team.

Kind regards,

on behalf of

Dr. Naoki Asano

Academic Editor

PLOS ONE